# Characterization and Functional Evaluation of *NK-lysin* from Clownfish (*Amphiprion ocellaris*)

Dapeng Yu [1,2,†] , Haohang Zhao [1,2,†], Yiming Wen [1,2], Tao Li [3], Hongli Xia [1,2], Zhiwen Wang [1,2], Zhen Gan [1,2], Liqun Xia [1,2], Jianlin Chen [1,2,*] and Yishan Lu [1,2,*]

1   Guangdong Provincial Engineering Research Center for Aquatic Animal Health Assessment, Shenzhen Public Service Platform for Evaluation of Marine Economic Animal Seedings, Shenzhen Institute of Guangdong Ocean University, Shenzhen 518000, China; yudapeng@gdou.edu.cn (D.Y.); zhaohaohang11@stu.gdou.edu.cn (H.Z.); wenyiming11@stu.gdou.edu.cn (Y.W.); xiahongli0427@163.com (H.X.); wangzhiwen@gdou.edu.cn (Z.W.); ganzhen258@gdou.edu.cn (Z.G.); xialq@gdou.edu.cn (L.X.)
2   Guangdong Provincial Key Laboratory of Pathogenic Biology and Epidemiology for Aquatic Economic Animals, College of Fisheries, Guangdong Ocean University, Zhanjiang 524000, China
3   Shenzhen Base of South China Sea Fisheries Research Institute, Chinese Academy of Fishery Sciences, Shenzhen 518000, China; 2112001133@gdou.edu.cn
*   Correspondence: jianlin-chen@m.scnu.edu.cn (J.C.); fishdis@163.com (Y.L.)
†   These authors contributed equally to this work and should be regarded as co-first authors.

**Abstract:** In previous studies, natural killer lysin (NK-lysin) emerged as a crucial antimicrobial peptide (AMP) discharged by NK cells and CTLs. The sequence of *NK-lysin* was cloned and discovered in some fishes, but its function remains unclear. In our study, we obtained a copy of *NK-lysin* from the spleen of the healthy clownfish (*Amphiprion ocellaris*; *AoNK-lysin*) through cloning and proceeded to investigate its potential functions and activities. The findings showed that the *AoNK-lysin* gene's open reading frame (ORF) had a length of 465 base pairs (bp) and encoded 154 amino acids (aa), which included a saposin B domain and six cysteine residues that are highly conserved, forming three intrachain disulfide bonds to carry out antimicrobial activity. The *AoNK-lysin* gene was widely present in different tissues, with the skin showing the highest expression, followed by the eye, intestine, and muscle. Additionally, the expression of *AoNK-lysin* was significantly upregulated in the immune organs (spleen, gill, intestine, and head kidney) of *A. ocellaris* after being challenged by Singapore group iridovirus (SGIV). Furthermore, a 399 base pair cDNA sequence that encodes the fully developed peptide of AoNK-lysin was successfully inserted into a secretion plasmid called pPIC9K. Subsequently, a significant amount of the recombinant AoNK-lysin protein was efficiently manufactured using the *Pichia pastoris* expression system. The antibacterial test demonstrated that the AoNK-lysin protein significantly suppressed the growth of various pathogens, particularly *Streptococcus agalactiae*, *Streptococcus iniae*, *Salmonella typhi*, *Shigella sonnei*, *Pseudomonas aeruginosa*, and *Aeromonas caviae*. The minimal inhibitory concentration (MIC) was found to be 7.81 μg/mL. Further analysis of antiviral assays showed all the viral mRNA of SGIV to be significantly reduced after AoNK-lysin protein stimuli in FHM cells. Collectively, these discoveries indicate that AoNK-lysin exhibits features of both direct pathogen-killing abilities and inhibited virus replication.

**Keywords:** *Amphiprion ocellaris*; NK-lysin; *Pichia pastoris*; antibacterial; antiviral

**Key Contribution:** The main finding of this study is that AoNK-lysin has the ability to eliminate pathogens and inhibited virus replication. This suggests that AoNK-lysin might act as an essential role in defense against bacteria and virus infection.



## 1. Introduction

The clownfish (*Amphiprion ocellaris*) is a well-known example of a fish that lives in anemones. It is a popular and valuable ornamental fish species that is commonly bred

and kept in both large aquariums and households [1]. The development of clownfish to aquaculture increased fishermen's income and declined the catching of wild clownfish in the natural sea area, as well as preserving coral reef biodiversity. However, clownfish aquaculture has suffered significant economic losses due to the presence of various pathogens, including bacteria (*Vibrio* sp., *Aliivibrio* sp., and *Bacillus* sp.), viruses (Alloherpes virus, Lymphocystis virus, and Singapore group iridovirus), and parasites (flagellates, monogeneans, amyloodinium, and cryptocaryon) [2]. In order to avoid these diseases in the commercial breeding of clownfish, antibiotics have been traditionally employed as a preventive measure against disease transmission. Nevertheless, the regular utilization of antibiotics gives rise to issues such as antibiotic-resistant bacteria, contamination of the environment, and risks to food safety [3]. Therefore, it is crucial to discover alternative approaches to hinder the occurrence of these illnesses in clownfish. Currently, antimicrobial peptides (AMPs) have garnered attention as a potential alternative to conventional antibiotics due to their effectiveness and wide-ranging ability to eliminate microorganisms. Additionally, they possess strong inhibitory properties against fungi, protozoa, and viruses [4]. Furthermore, the antibacterial action of AMPs is clearly distinct from that of antibiotics, and AMPs have a lower likelihood of inducing bacterial resistance.

Antimicrobial peptides (AMPs), initially identified in *Hyalophora cecropia* [5] and *Xenopus laevis* [6], play a crucial role in the immune system of all organisms by fighting against harmful microorganisms [7]. NK-lysin, a constituent of antimicrobial peptides (AMPs), is primarily synthesized by natural killer cells (NK cells) and cytotoxic T lymphocytes (CTLs) in mammals [8] and various teleost fish [9]. It possesses a surfactant-associated protein B domain that includes six highly conserved cysteine residues, forming three stable disulfide bridges essential for its antimicrobial function [10]. On the other hand, the teleost fish NK-lysin exhibits evident bactericidal properties. For instance, the NK-lysin found in Nile tilapia (*Oreochromis niloticus*) and grouper (*Epinephelus coioides*) effectively kills *Aeromonas hydrophila* and *Proteus mirabilis* at a concentration of 7.81 μg/mL, as demonstrated by minimal inhibitory concentration tests [11]. Furthermore, apart from its antimicrobial properties, teleost fish NK-lysin also exhibits immunomodulatory effects [12]. For instance, the in vivo assay of half-smooth tongue sole demonstrated the induction of il-1β and il-8 genes by *NK-lysin*, while the in vitro assay of Atlantic salmon showed similar results [13]. Moreover, the use of NK-lysin through intraperitoneal injection has been shown to enhance the survival of fish when faced with pathogenic infections in various fish types. For instance, it has been observed to be effective against *Aeromonas hydrophila* infection in barbel steed (*Hemibarbus labeo*) [14], nodavirus infection in European sea bass (*Dicentrarchus labrax*) [15], and *Philasterides dicentrarchi* infection in turbot (*Scophthalmus maximus*) [16]. NK-lysin's remarkable capacity to directly eliminate microbes and modulate the immune system makes it a promising alternative to antibiotics for fighting pathogenic microbial infections in teleost fish.

Recombinant DNA technology is considered a cost-effective method to produce large quantities of active AMPs, which is very much required for industrial applications [17]. The methylotrophic yeast *Pichia pastoris*, being a eukaryotic expression system, has been extensively utilized for expressing heterologous peptides with post-translational modifications to ensure their complete functionality [18]. Therefore, our objective was to replicate and examine the *NK-lysin* gene of clownfish (*Amphiprion ocellaris*; *AoNK-lysin*). Subsequently, the genetic code for a fully developed peptide of AoNK-lysin was introduced into the eukaryotic expression plasmid Ppic9K. This plasmid was then transferred into the *Pichia pastoris* GS115 strain in order to produce a substantial amount of recombinant protein. The purpose of this protein production was to investigate its potential in combating pathogens and enhancing the immune system. This study might provide an insight in further application on AoNK-lysin as immunopotentiator to feed fish against bacterial and viral infection.

## 2. Materials and Methods

### 2.1. Ethics Statement

The clownfish experiment followed the regulations of animal testing at Guangdong Ocean University, and the animal facility adhered to the guidelines for lab care and use set by the National Institutes of Health (NIH Publications No. 8023).

### 2.2. Fish, Cell Lines, Virus, and Bacteria

The Shenzhen Base of South China Sea Fisheries Research Institute in Guangdong Province, China, bred and maintained clownfish (*A. ocellaris*) in their sterile seawater aquaculture filtration system prior to conducting additional experiments. Meanwhile, the clownfish were fed twice daily with normal aquatic feed. Epithelial cells of the fathead minnow (FHM) were kept in a laboratory tank of liquid nitrogen ($-156\,^{\circ}$C) until they were ready for use [19]. The Singapore grouper iridovirus (SGIV) was kindly delivered by the schoolfellow senior engineer of fisheries, Mr. Song (Zhongshan, China). The strain (GS115) expressing yeast was bought from Thermo-Fisher Scientific Inc. (Invitrogen, Carlsbad, CA, USA) and grown in yeast peptone dextrose (YPD) liquid medium at a temperature of $30\,^{\circ}$C until it was ready for use. Total RNA from clownfish tissues was treated with TRIzol reagent (TransGen, Beijing, China) according to the manufacturer's protocol. Our lab received the *Aeromonas hydrophila strain* BYK00810 from the Key Laboratory of Exploration and Utilization of Aquatic Genetic Resources, Ministry of Education (Shanghai, China), for which we are grateful. The supplement used in this experiment was obtained from the China Institute of Veterinary Drug Control (Beijing, China).

### 2.3. Cloning the Open Reading Frame (ORF) of AoNK-lysin

Total RNA was extracted from various tissues (skin, muscle, eye, intestine, brain, and so on) of clownfish and followed by cDNA synthesis (TransGen, Beijing, China) according to the manufacturer's protocol, as previously described [9]. In short, the initial quality and quantity of each RNA sample were checked via an Agilent 2100 Bioanalyzer (Agilent, Santa Clara, CA, USA), and their integrities were examined by electrophoresis on 0.8% RNase-free agarose gel. Samples were diluted and RNA concentration was measured via NanoDrop 2000 (Thermo Scientific, Waltham, MA, USA). The same equipment was used to assess purity through the OD260/OD230 nm (2.0–2.4) and OD260/OD280 nm (1.8–2.0) absorbance ratios. The RNA samples with an RNA integrity number (RIN) more than 8.0 were considered acceptable for subsequent cDNA synthesis via TransScript One-Step gDNA Removal and cDNA Synthesis SuperMix (TransGen, Beijing, China) according to the manufacture instruction. Based on the published genome information of clownfish in the NCBI library, the specific primers AoNK-OF/AoNK-OR were used for cloning the ORF sequence of *AoNK-lysin* (XM-023284512.1) (Table 1).

**Table 1.** Primers for cloning the *AoNK-lysin* gene and quantitative real-time PCR (qRT-PCR) analysis.

| Primer Name | Sequence (5′−3′) | Purpose |
|---|---|---|
| AoNK-OF | ATGGAAAGAATTTCAATCCTG | Amplification of ORF of *AoNK-lysin* |
| AoNK-OR | TCCATGCATAGGATGTACAG | |
| AoNK-MF | CCGG<u>GAATTC</u>AGAAACATAGAGGTCAGCATCAGTGA | Amplification of sequence of mature *AoNK-lysin* |
| AoNK-MR | AAGGAAAAAA<u>GCGGCCGC</u>TTACATCATCACCATCA-CCATTCCATGCATAGGATGTACAGGA | |
| 5′ AOX1 | GACTGGTTCCAATTGACAAGC | Identification of integration in pPic9K |
| 3′ AOX1 | GCAAATGGCATTCTGACATCC | |
| SGIV-F | AGAGTTTTCGGTCGGGGTTC | Amplification of sequence of SGIV MCP |
| SGIV-R | GAAACGAGACCCACGGTCAT | |
| β-actin-F | GGGCCAAAAGGACAGCTAC | |
| β-actin-R | CAGGGTCAGGATACCCCTCT | qRT-PCR in *AoNK-lysin* |
| qNKlysin-F | TGGAGGAGATGGACACGGAT | |
| qNKlysin-R | TGGAGGAGATGGACACGGAT | |

**Table 1.** *Cont.*

| Primer Name | Sequence (5′−3′) | Purpose |
|---|---|---|
| qORF115-F | CGGAAAGAACACAGATAACGG | |
| qORF115-R | AAAAAACACATGGCTTGCAAA | |
| qORF072-F | GCACGCTTCTCTCACCTTCA | |
| qORF072-R | AACGGCAACGGGAGCACTA | |
| qORF049-F | ATGTACGTATACCCCGCAAT | |
| qORF049-R | TCATTTTTTTTGCCTAA | qRT-PCR in SGIV |
| qORF086-F | ATCGGATCTACGTGGTTGG | |
| qORF086-R | CCGTCGTCGGTGTCTATTC | |
| qActin-F | TACGAGCTGCCTGACGGACA | |
| qActin-R | GGCTGTGATCTCCTTCTGCA | |

### 2.4. Sequence Analysis of the AoNK-lysin ORF Gene

The methods of sequence analysis were performed as previously described [9]. The cDNA sequence and amino acid sequence of the AoNKlysin similarities were examined via BLAST (http://blast.ncbi.nlm.nih.gov/Blast.cgi (accessed on 6 January 2023)). The physical and chemical properties were predicted using ExPASy software 2.0 (http://www.expasy.org/ (accessed on 6 January 2023)). The location of domains was predicted by SMART (http://smart.embl-heidelberg.de/ (accessed on 6 January 2023)). Protein family membership was predicted by the InterProScan program (http://www.ebi.ac.uk/Tools/pfa/iprscan/ (accessed on 6 January 2023)). Protein multiple sequence alignments of the AoNKlysin protein was performed by the ClustalX 2.0 program with the default parameters and edited by the GeneDoc software. The phylogenetic tree was generated based on the deduced amino acid sequence of the AoNKlysin protein with the neighbor-joining method using MEGA 6 program, in which the Poisson distribution substitution model and bootstrapping procedure with 1000 bootstraps were applied.

### 2.5. Distribution Expression of AoNK-lysin

To detect the expression of the *AoNK-lysin* gene in various tissues, a total of 9 diverse tissue samples were gathered from healthy clownfish and preserved using RNAlater. Section 2.3 involved the extraction of total RNA and the synthesis of cDNA from these samples. The qTOWER3G Real-Time PCR machine (Analytik Jena, Jena, Germany) was used to perform the qRT-PCR. The primers qNKlysin-F/qNKlysin-R for qRT-PCR were created based on the *AoNK-lysin* gene and can be found in Table 1. The qRT-PCR process was conducted in the following manner: initially, preheat at 95 °C for 5 min, followed by 40 cycles of 95 °C for 15 s and 60 °C for 30 s. The clownfish β-actin gene was utilized as an internal control to standardize the data. The investigation of the *AoNK-lysin* gene transcription was conducted utilizing the $2^{-\Delta\Delta Ct}$ technique.

### 2.6. Clownfish Challenged after SGIV Infection

Before the SGIV infection experiment, 100 healthy clownfish were randomly averaged into 2 groups (50 fish/group). The clownfish were intraperitoneally (i.p.). The control group was given the same volume of M199 medium, while the experimental group was administered with 10 μL SGIV solution (resuspended in M199 medium, MOI = 1). Specimens (gill, head kidney, spleen, and intestine) were collected at 0, 6, 12, 24, 48, and 72 h after SGIV infection. The extraction of total RNA from each sample was carried out, and the cDNA was transcribed in reverse, as described in Section 2.3. Furthermore, the results of qRT-PCR were analyzed as in Section 2.5.

### 2.7. Plasmid Construction

The mature *AoNK-lysin* sequence was cloned using the specific primers AoNK-MF/AoNK-MR, which contained *EcoR I* and *Not I* enzyme cleavage sites. The cloned sequence was then combined with the pMD-18T plasmid from TaKaRa, Japan. Subsequently, the recombi-

nant pMD-AoNK-lysin plasmid was successfully transferred to Trans 5α *Escherichia coli* (TransGen, Beijing, China) and underwent accurate sequencing to facilitate subsequent investigations. Both the recombinant pMD-AoNK-lysin and pPIC9K plasmids were digested at the same time using the *EcoR* I and *Not* I enzymes. Subsequently, the *AoNK-lysin* segment was integrated into pPIC9K, resulting in the acquisition of the recombinant pPIC9K-AoNK-lysin plasmid expressed in yeast. Then, the recombinant pPIC9K-AoNK-lysin plasmid was successfully transferred to Trans 5α *E. coli* (TransGen, Beijing, China) and underwent accurate sequencing for subsequent investigations. As per the guidelines of the Plasmid Mini Kit (Promega, Madison, WI, USA), the plasmid pPIC9K-AoNK-lysin was acquired using the kit and kept at a temperature of −20 °C until it was needed.

### 2.8. Yeast Expression and Purification of the AoNK-lysin Protein

pPIC9K-AoNK-lysin was digested with the *Sal I* enzyme (NEB) and subsequently converted to the *P. pastoris GS115* strain. The primers 5′AOX1/3′AOX1 (Table 1) were used to identify positive clones of GS115/pPIC9K-AoNK-lysin and confirm the presence of the *AoNK-lysin* gene in *P. pastoris* strain GS115. In accordance with the instructions provided in Chen's publication [9], the process of expressing and purifying recombinant AoNK-lysin in *P. pastoris* using yeast was carried out.

### 2.9. Antibacterial Function of AoNK-lysin

The MIC experiment was carried out following the previously described methods [9]. In summary, every single bacterium was cultivated until reaching the typical logarithmic stage and then suspended in PBS at a concentration of $1 \times 10^5$ CFU/mL. The AoNK-lysin protein was diluted with PBS buffer in a twofold manner, and the kanamycin (Kana$^+$) (100 μg/mL) was used as the positive control group, as previously mentioned in [9]. Bacteria were cultivated in 96-well polypropylene microtiter plates, where 100 μL of the microorganisms were cultured and combined with the serially diluted AoNK-lysin and Kana$^+$. The plates, which contained the mixed medium, were subsequently grown at 30 °C over 18 h. Based on MIC measurements, the lowest level of AoNK-lysin protein that prevented the growth of bacteria was determined. The experiment was replicated three times.

### 2.10. Effect of the AoNK-lysin Protein on Viral Gene Transcription

Before conducting the SGIV infection experiment, FHM cells were cultivated on 24-well polypropylene microtiter plates at a temperature of 25 °C for a duration of 24 h. The cells were incubated with the AoNK-lysin protein at concentrations of 0 and 10 μg/mL at 25 °C for 24 h. After that, the medium containing SGIV was replaced, and the cells were incubated with SGIV for another 24 h. Finally, the cells were harvested for further study. qRT-PCR was used to investigate the expression of the viral gene of SGIV in order to explore the existence of the virus.

### 2.11. Statistical Analysis

The data are displayed as the average ± SEM (standard error of the mean). The data were edited using GraphPad Prism software, and statistical analysis was conducted using one-way ANOVA with SPSS statistics 25.0 software. The data represent the averages of three separate trials, with statistical significance indicated by asterisks in the figures as follows: $p > 0.05$, indicating no significance; $p < 0.05$ (*), indicating significance; and $p < 0.01$ (**), indicating high significance.

## 3. Results

### 3.1. Sequence Analysis of AoNK-lysin

The spleen cDNA successfully cloned the ORF sequence of *AoNK-lysin* (465 bp), resulting in a putative 154 aa protein with an estimated molecular mass of 17.1 kDa and a theoretical isoelectric point (PI) of 6.71 (Figure 1A). According to the figure, it has been anticipated that *AoNK-lysin* possesses a signal peptide (located at amino acids 1–20) and a surfactant-associated protein B domain (amino acids 50–122) at the C-terminus (Figure 1B). The SWISS-Model predicted the tertiary structures of the AoNK-lysin protein in humans, zebrafish, and clownfish. The outcome indicated a strong resemblance between AoNK-lysin and zebrafish NK-lysin's tertiary structures (Figure 1C). The analysis of synteny showed that clownfish and other fish NK-lysin counterparts were situated in the identical position next to mtfr1l and tonsl (Figure 1D) and displayed a typical saposin B domain. Figure 2 shows high similarity in the six conserved cysteine residues among various species, as determined by multiple sequence alignment. Moreover, the multiple sequence alignment of the NK-lysin protein with various species exhibited identity ranging from 21.74% to 89.24%. According to the phylogenetic tree, AoNK-lysin was grouped with teleost fish and showed the closest resemblance to *A. polyacanthus* NK-lysin (Figure 3).

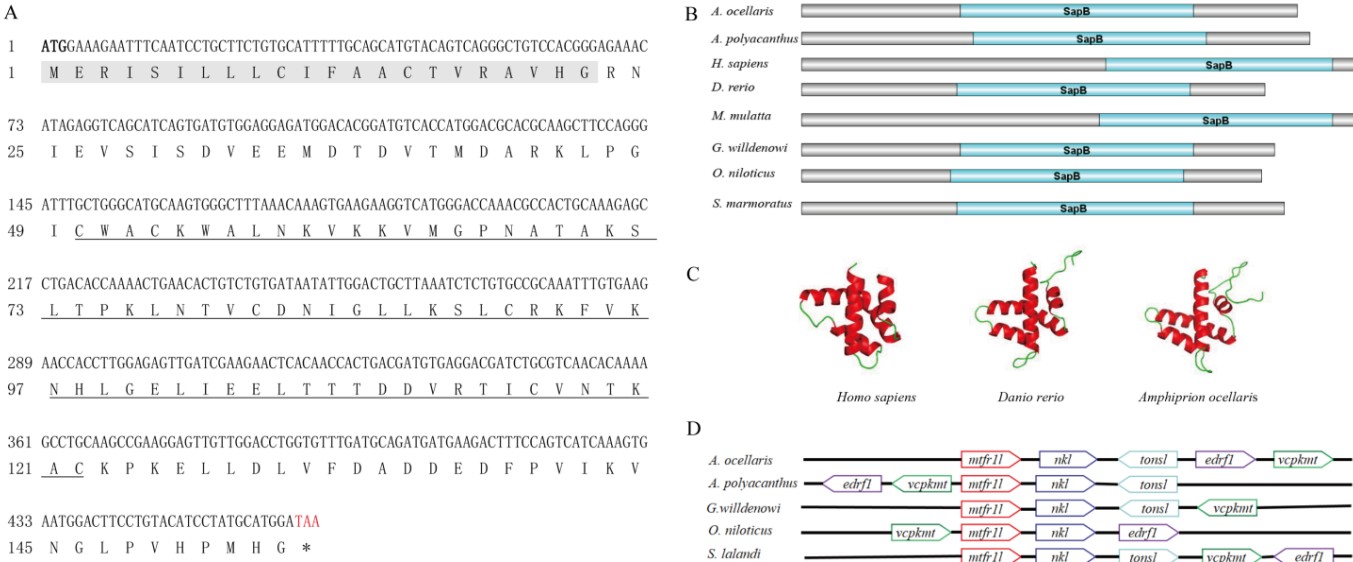

**Figure 1.** Sequence characteristics of *NK-lysin* of *Amphiprion ocellaris*. (**A**) Sequence analysis of *AoNK-lysin*. The bold font represents translation start codon. The red font represents translation stop codon. The gray shaded part represents signal peptide. The underline part represents saposin B domain. (**B**) Domain predication of NK-lysin in *Amphiprion ocellaris*, *Acanthochromis polyacanthus*, *Homo sapiens*, *Danio rerio*, *Macaca mulatta*, *Gouania willdenowi*, *Oreochromis niloticus*, and *Sebastiscus marmoratus*. The blue part represents the conservative domain of SapB. (**C**) A 3D structure of the NK-lysin protein was modeled using the Swiss-model database and visualized with PyMOL software. The α-helices and random coils in the models were colored blue and purple, respectively. (**D**) The gene collinearity analysis of the *NK-lysin* genes from *Amphiprion ocellaris*, *Oreochromis niloticus*, *Acanthochromis polyacanthus*, *Dicentrarchus labrax*, and *Siniperca chuatsi*. The direction of the arrows indicates the orientation of gene transcription.

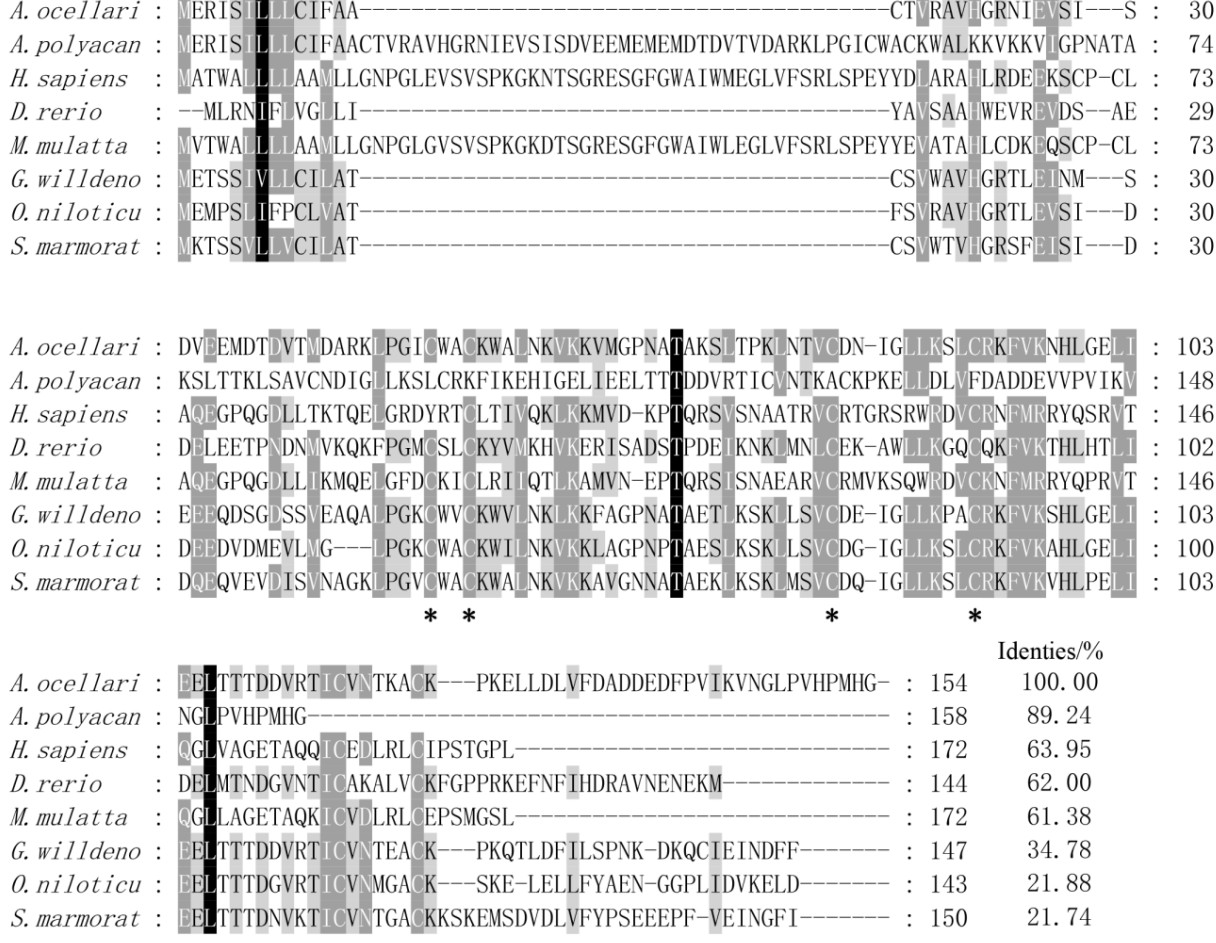

**Figure 2.** Multiple alignment of the deduced amino acid sequences of protein NK-lysin among different species. The * indicates the cysteine. GenBank accession numbers are shown in Figure 3.

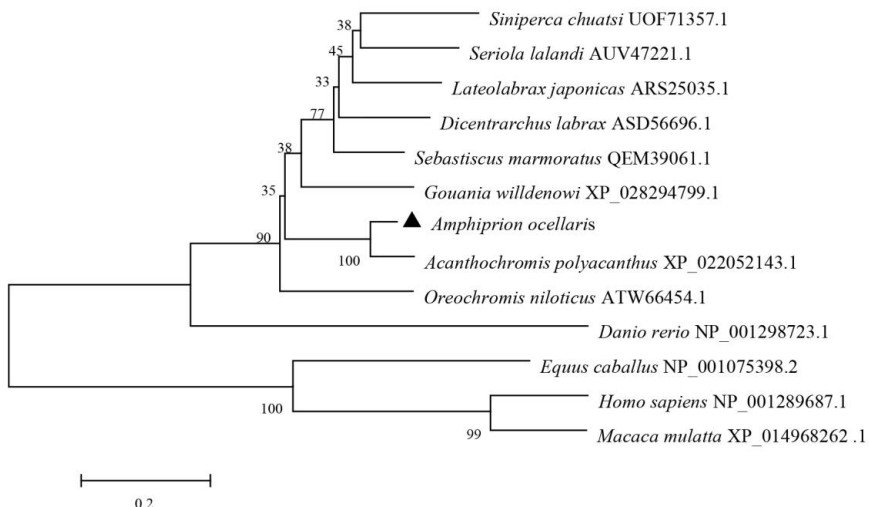

**Figure 3.** Phylogenetic tree of NK-lysin family members constructed using the neighbor-joining method. The numbers in each branch indicate percentage bootstrap values for 1000 replicates. The triangle represents Clownfish.

*3.2. The Expressions of AoNK-lysin in Clownfish*

The presence of the *AoNK-lysin* gene was examined in nine different tissues obtained from nine healthy fish. As shown in Figure 4A, the *AoNK-lysin* gene showed extensive expression in all the examined tissues, with the greatest levels observed in the skin, eye, and intestine. It exhibited moderate expression in the muscle, spleen, brain, and head kidney, while its expression was low in the gill and liver. This result is different from the previous finding, especially mammalian [10]. In mammalian organisms, the *NK-lysin* gene showed increased expression in immune tissues, such as lymphoid tissues and cells [20,21]. On the other hand, the *NK-lysin* gene in teleosts exhibited significant expression in the gill [22], head kidney [23], intestine [24], and spleen [25], and it plays a crucial function in acquired immune protection and the response to inflammation. Based on these observations, our study revealed that the expression of the *AoNK-lysin* gene was highest in the eye and skin. The unique expression pattern of the *AoNK-lysin* gene could be associated with the habitat of clownfish, which live in close proximity to sea anemones. This association may contribute to the induction of *AoNK-lysin* expression in the eyes and skin, possibly influenced by the presence of sea anemones.

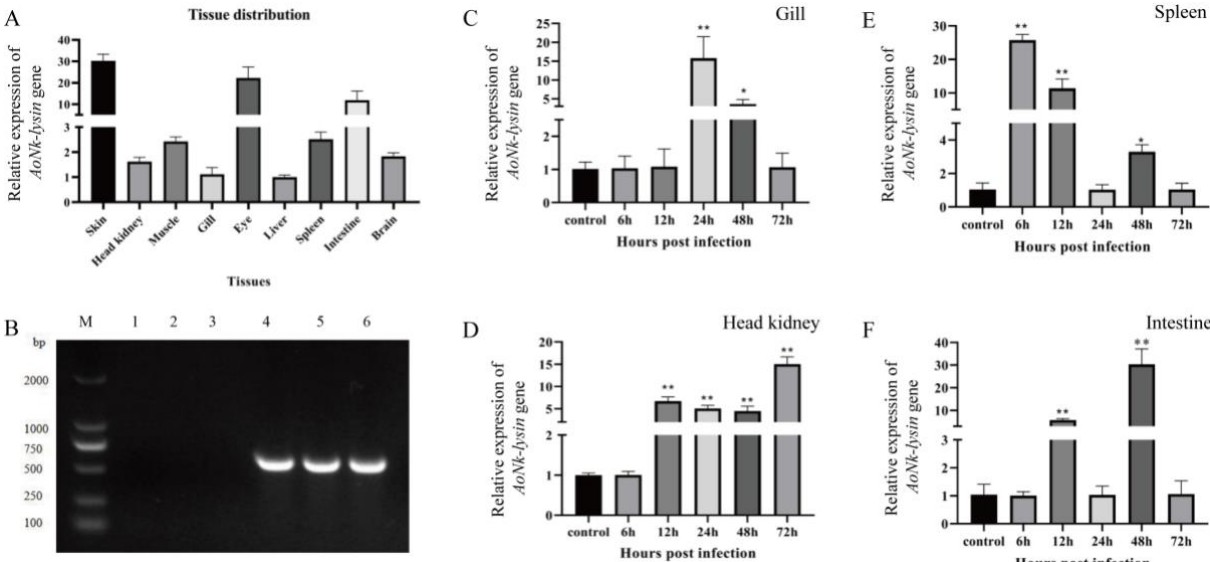

**Figure 4.** Fish were intraperitoneally injected with 10 μL SGIV (MOI = 1) or an equal volume of L15 medium. (**A**) Tissue distribution of the *AoNK-lysin* gene measured by quantitative real-time PCR (qRT-PCR). (**B**) SGIV was identified by PCR; lanes 1–3 were injected with L15; lanes 4–6 were injected with SGIV. (**C–F**) Expression of the *AoNK-lysin* gene was normalized against *β-actin* gene expression. The temporal expression profile of the *AoNK-lysin* gene of clownfish after SGIV infection. The mRNA expression of the *AoNK-lysin* gene in peripheral gill, head kidney, spleen, and intestine after injection with SGIV for 6, 12, 24, 48, and 72 h. Data are shown as mean $\pm$ SEM ($n$ = 4). * $p < 0.05$ or ** $p < 0.01$ is considered significant.

In order to guarantee the effective transmission of SGIV to clownfish, we utilized SGIV-MCP primers to clone and identify the primary capsid protein of SGIV that was obtained from the spleen after a 6 h infection (Figure 4B). In order to investigate the expression pattern of *AoNK-lysin* during viral infection, clownfish were intraperitoneally injected with SGIV at a multiplicity of infection (MOI) of 1. After SGIV infection (Figure 4C–F), the expression of the *AoNK-lysin* gene showed a significant increase in the gill, intestine, head kidney, and spleen. The *AoNK-lysin* gene reached its highest point after 24 h in gill, while in the spleen, it peaked after 6 h. Notably, the expression of the *AoNK-lysin* gene was notably increased at 12 h in the intestine and head kidney.

*3.3. Recombinant AoNK-lysin Protein Expression and Antibacterial Activity*

After successfully converting pPIC9K-NK-lysin into the GS115 strains, we proceeded to sequence and identify the positively expressing clone, which will be utilized for subsequent investigations. SDS-PAGE analysis confirmed the production of the mature AoNK-lysin protein by *P. pastoris*, and its actual peptide size matched the predicted size of the mature protein (14.5 kDa), as shown in Figure 5A. Furthermore, the mature AoNK-lysin recombinant proteins were confirmed through Western blot analysis, employing the monoclonal anti-His-tag antibody (Figure 5B). Afterwards, the fully developed AoNK-lysin protein was obtained and gathered using a protein purification kit, resulting in a yield of 1 mg/mL. The antimicrobial activity was assessed using a MIC assay to determine its biological function. The effective range of the recombinant AoNK-lysin protein against both Gram-positive and Gram-negative bacteria was discovered. Among them, the minimum inhibitory concentration (MIC) of AoNK-lysin protein against *S. agalactiae*, *S. iniae*, *S. typhi*, *S. sonnei*, *P. aeruginosa*, and *A. caviae* was 7.81 μg/mL. For *B. subtilis* and *P. mirabilis*, the MIC was 15.63 μg/mL. The MIC for *E. tarda*, *K. pnenmoniae*, and *A. hydrophila* was 31.25 μg/mL. As for *S. aureus* and *S. typhimurium*, the MIC was 125 μg/mL. Nevertheless, the recombinant AoNK-lysin protein exhibited no antibacterial activity against other tested bacteria, including *E. coli*, *V. cholerae*, and *P. vulgari*, with minimum inhibitory concentrations (MIC) exceeding 250 μg/mL (Table 2).

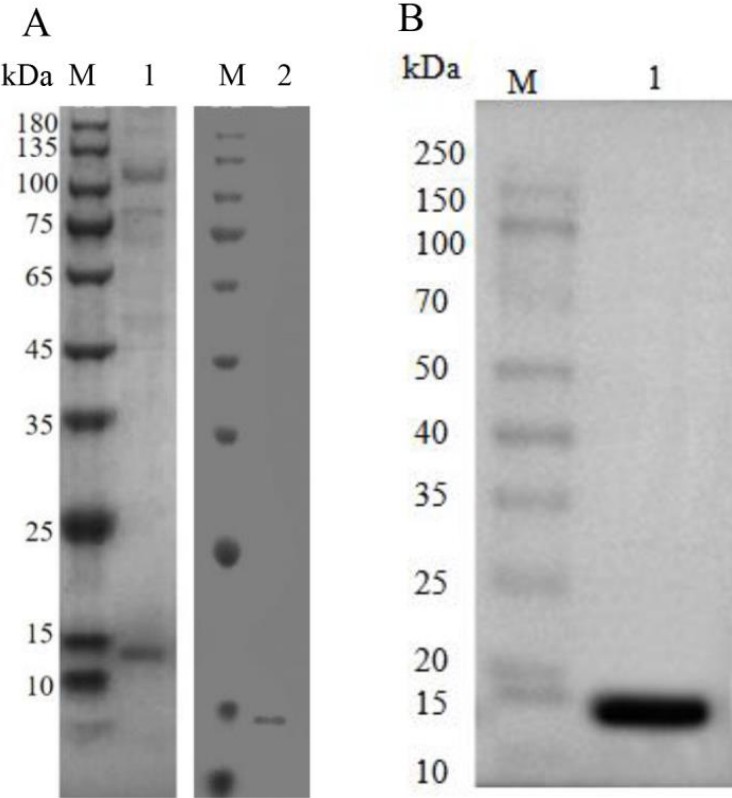

**Figure 5.** Expression and purification of recombinant protein of mature AoNK-lysin by SDS-PAGE analysis. Lane M indicates protein with standard molecular masses. (**A**) Lane 1 indicates expressed recombinant protein of mature AoNK-lysin. Lane 2 indicates purified recombinant protein of mature AoNK-lysin. (**B**) Western blotting analysis of recombinant protein AoNK-lysin (lane 1) by using antibody against the His-tag.

**Table 2.** Minimal inhibitory concentration of AoNK-lysin protein against a panel of microorganisms.

| Bacterial Strains | | Minimal Inhibitory Concentration (MIC; µg/mL) | |
|---|---|---|---|
| | | AoNK-lysin | Kana⁺ |
| Gram-positive bacteria | *Staphylococcus aureus* | 125 | + |
| | *Bacillus subtilis* | 15.63 | + |
| | *Streptococcus agalactiae* | 7.81 | + |
| | *Streptococcus iniae* | 7.81 | + |
| Gram-negative bacteria | *Edwardsiella tarda* | 31.25 | + |
| | *Escherichia coli* | >250 | + |
| | *Vibrio cholerae* | >250 | + |
| | *Klebsiella pnenmoniae* | 31.25 | + |
| | *Salmonella typhi* | 7.81 | + |
| | *Aeromonas hydrophila* | 31.25 | + |
| | *Shigella sonnei* | 7.81 | + |
| | *Pseudomonas aeruginosa* | 7.81 | + |
| | *Aeromonas caviae* | 7.81 | + |
| | *Proteus vulgaris* | >250 | + |
| | *Salmonella typhimurium* | 125 | + |
| | *Proteus mirabilis* | 15.63 | + |

*3.4. Bioactivity Analysis of AoNK-lysin Protein*

In order to investigate the role of the AoNK-lysin protein in viral replication, we chose the viral genes ORF 049, ORF 072, ORF086, and ORF115 of SGIV to analyze viral gene transcription using qRT-PCR. As shown in Figure 6, after stimulating the AoNK-lysin protein at an optimal concentration of 10 µg/mL for 24 h, the transcription of all viral genes was markedly decreased compared to the control.

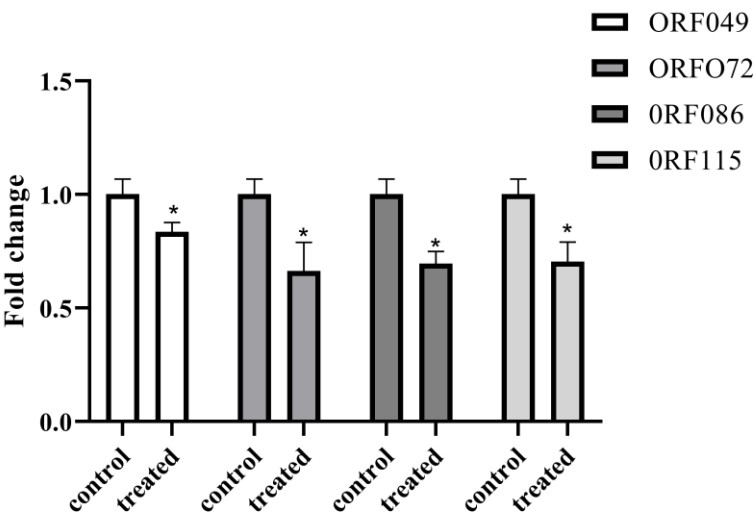

**Figure 6.** qRT-PCR analysis of viral genes ORF 049, ORF 072, ORF 086, and ORF 115 transcriptions stimulated with lysin for 24 h during SGIV infection. Data are shown as mean ± SEM ($n = 4$). * $p < 0.05$.

**4. Discussion**

In this study, a novel *NK-lysin* gene was cloned from *A. ocellaris*, *AoNK-lysin*, and we identified its biological function, structure, and expression pattern in grouper after stimulation. Our study showed that *AoNK-lysin* belonged to the saposin-like protein family. And members of the saposin-like protein family have a notable characteristic in common, which is the existence of six cysteines positioned similarly and creating three intrachain disulfide linkages [12]. The current investigation revealed that the AoNK-lysin contains a saposin B region and six cysteine compounds. The cysteines are situated in positions that

correspond to the cysteines in porcine NK-lysin that form disulfide bonds. It was predicted that these cysteines would form three disulfide links, using the same cysteine pairing as in mammalian NK-lysin. The alignment of sequences indicated that the six cysteine residues are greatly preserved in teleost fish, indicating that the three disulfide bonds created by these cysteines might play a crucial role in the biological function of NK-lysin in lower vertebrates, similar to what is observed in higher vertebrates.

In this study, it was found that the expression of *NK-lysin* in *A. ocellaris* was different from the previous finding, especially in mammals [10]. In mammalian organisms, the *NK-lysin* gene showed increased expression in immune tissues, such as lymphoid tissues and cells [20,21]. On the other hand, the *NK-lysin* gene in teleosts exhibited significant expression in the gill [22], head kidney [23], intestine [24], and spleen [25], and it plays a crucial function in acquired immune protection and the response to inflammation. Based on these observations, our study revealed that the expression of the *AoNK-lysin* gene was highest in the eye and skin. The highest expression of *AoNK-lysin* in skin was different among the *NK-lysin* expression in other fishes, while differences among fish species may be considered, especially for the genetic differences. On the other hand, the habitat and living environment may also cause this type of difference. The unique expression pattern of the *AoNK-lysin* gene could be associated with the habitat of clownfish, which live in close proximity to sea anemones. This association may contribute to the induction of *AoNK-lysin* expression in the eyes and skin, possibly influenced by the presence of sea anemones.

After viral infection, comparable patterns of *NK-lysin* gene expression were documented in other investigations on teleosts. After being exposed to megalocytivirus [23], the head kidney and spleen showed a significant increase in the expression of NK-lysin in tongue sole (*Cynoglossus semilaevis*). After being challenged with SGIV [11], the head kidney, spleen, gill, and intestine of grouper (*Epinephelus coioides*) exhibited a significant increase in the expression level of *NK-lysin*. The findings indicated that teleost *NK-lysin* might play a role in protecting the immune system after viral infections.

Previous studies have shown that over 10 artificial polypeptides with antibacterial properties were obtained from mammalian NK-lysin [4]. Furthermore, a range of NK-lysin peptides was found in marine creatures, exhibiting a bactericidal impact on microorganisms in laboratory settings [22,24,26–28]. In the same way, the current study demonstrated that the NK-lysin exhibited a wide-ranging antibacterial activity against Gram-positive bacteria (*S. aureus, B. subtilis, S. agalactiae,* and *S. iniae*) as well as Gram-negative bacteria (*S. aureus, B. subtilis, S. agalactiae, S. iniae, E. tarda, K. pnenmoniae, S. typhi, A. hydrophila, S. sonnei, P. aeruginosa, A. caviae, S. typhimurium,* and *P. mirabilis*). NK-lysin peptide effectively eradicated the Gram-negative bacteria in contrast to the Gram-positive bacteria, possibly due to variations in the composition of their cell surfaces. The NK-lysin peptide has a good effect in killing various microorganisms in vitro. However, the widespread use and application of costly polypeptide production via chemical synthesis is restricted. In order to substitute the chemical synthesis technique, it was necessary for us to develop a fresh approach for manufacturing NK-lysin. Using the yeast expression system, we obtained a large quantity of AoNK-lysin protein in the current investigation. In summary, the *P. pastoris* GS115 strain, harboring a fully developed *AoNK-lysin* gene, holds promising prospects for future commercial and practical applications.

SGIV is an extremely infectious fish virus that leads to significant death rates and substantial financial damage. Similar to other viruses, SGIV requires entry into live cells in order to reproduce [29]. Virus replication heavily relies on the significant involvement of the major capsid protein (MCP). The SGIV major capsid protein (MCP) [30,31] includes the ORF072 gene and ORF049 gene. The SGIV virus encodes the ICP18 ortholog gene known as ORF086, which is involved in SGIV replication and maturation [30]. After viral infection, the expression of immune-related genes was decreased by the ORF115 gene, which was encoded by SGIV, and the IRF-3 gene was also activated [32]. It is common knowledge that the manifestation of the virus MCP gene signifies the advancement of viral infiltration into cells. In this study, we examined the expression patterns of ORF049, ORF072, ORF086, and

ORF115 following viral infection. The qRT-PCR findings indicated a significant decrease in the expression of these genes, suggesting that the cells inhibited SGIV replication after AoNK-lysin incubation.

## 5. Conclusions

To summarize, the open reading frame (ORF) of the *AoNK-lysin* gene was 465 base pairs (bp), and it encoded 154 amino acid (aa) residues, which includes a saposin B domain and six cysteine residues that are highly conserved. These cysteine residues are responsible for forming three intrachain disulfide bonds, which are crucial for the antimicrobial activity of the gene. Furthermore, the *AoNK-lysin* gene exhibited constant expression in different tissues, with the skin showing the highest level of expression. Moreover, its expression was notably increased in the gill, intestine, head kidney, and spleen of *A. ocellaris* after challenged by SGIV. In addition, we successfully developed the GS115/pPIC9K-AoNK recombinant strain of *P. pastoris*, which is capable of effectively producing the AoNK-lysin antimicrobial peptide derived from *A. ocellaris*. The antimicrobial properties indicated that the AoNK-lysin protein has the ability to effectively suppress the proliferation of various harmful bacteria and the replication of SGIV. The distinctive characteristics indicate that AoNK-lysin could serve as a potential replacement for antibiotics, and the *P. pastoris* expression system has the potential for producing AoNK-lysin protein on a large scale as an immunopotentiator to prevent and treat diseases in clownfish and other fish species in the future.

**Author Contributions:** D.Y.: writing—original draft, writing—review and editing. Y.W.: data curation, formal analysis, investigation, writing—review and editing, data curation. H.Z.: investigation, data curation. T.L.: formal analysis, investigation. H.X.: investigation, data curation. Z.W.: formal analysis, investigation. Z.G.: formal analysis, investigation. L.X.: project administration, validation, resources, funding acquisition, writing—review and editing. J.C. and Y.L.: conceptualization, validation, funding acquisition, project administration, resources, writing—review and editing. All authors have read and agreed to the published version of the manuscript.

**Funding:** This work was supported by the National Key Research and Development Project (2020YFD0900201), National Natural Science Foundation of China (32102827, 31972818, 31472302), China Postdoctoral Science Foundation (2019M662959), Guangdong Basic and Applied Basic Research Foundation (2019A1515110987), Zhongshan Social Welfare and Basic Research Project (210803114048631, 2108031148611), and project supported by the Shenzhen Science and Technology Program (JCYJ202103324130014035). This work is supported by special funds for science technology innovation and the Shenzhen Industrial Development Special Fund Project (Project No: 1303).

**Institutional Review Board Statement:** The animal study protocol was approved by the Ethics Committee of the Guangdong Ocean University (protocol code (2019) 1 and date of approval 10 May 2019).

**Data Availability Statement:** Data available on request from the authors. The data that support the findings of this study are available from the corresponding author, [author J.C.], upon reasonable request.

**Conflicts of Interest:** All the authors declare that they have no known competing financial interests or personal relationships that could have appeared to influence the work reported in this paper.

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
