# Peer review of "Characterization and Functional Evaluation of NK-lysin from Clownfish (Amphiprion ocellaris)"

_fishes, doi:10.3390/fishes8110533_

Round 1

Reviewer 1 Report

I believe the manuscript has

been significantly improved and now warrants publication in fishes

Author Response

Response to reviewer.

Comments to the Author (in italics) with our responses (bold script)

We believe that we have accommodated the requests of the reviewer and in doing so, the quality of the manuscript has been greatly improved.

Reviewer 1

I believe the manuscript has been significantly improved and now warrants publication in fishes.

Thanks for your review.

Reviewer 2 Report

The whole manuscript needs critical English editing and revising the typos.

Abstract: L 15-17 why are you discussing other studies in the abstract?

-summarise the results included in the abstract section.

Introduction:

L 46-50: Revise and rewrite.

M and M:

L 103: How many fishes.

L 113: Clarify total RNA extraction.

L 119: "and cDNA synthesis was performed previously" what does this mean??

Table 1: -You need to leave a pace between the primer name and F or R.

- Correct "Used" in the main raw.

- Mention the source of primers.

L 125-127: Describe more details, it is not enough to just cite your previous work and what does SUPPLEMENT MEAN???

L 129: "A total of 9 129 diverse samples were gathered" What does this mean??

L 141: "  averaged into 2" WHAT IS THIS??

Discussion:

- better to separate results from the discussion.

- Try to make a better correlation between your findings.

It requires extensive Revision and editing. 

Author Response

Response to reviewer.

Comments to the Author (in italics) with our responses (bold script)

We believe that we have accommodated the requests of the reviewer and in doing so, the quality of the manuscript has been greatly improved.

Reviewer 2

The whole manuscript needs critical English editing and revising the typos.

We appreciate the valuable suggestion. The manuscript has been carefully checked and reviewed by a native English speaker.

Abstract: L 15-17 why are you discussing other studies in the abstract? -summarise the results included in the abstract section.

We appreciate the valuable comment and had deleted the description on discussing other studies in the abstract section in the revised manuscript.

Introduction:

L 46-50: Revise and rewrite.

We appreciate the helpful reviewer’s comment and rephrase the sentence as follow in the revised manuscript:.

It is a popular and valuable ornamental fish species that is commonly bred and kept in both large aquariums and households [1].(L 42-43) However, clownfish aquaculture has suffered significant economic losses due to the presence of various pathogens, including bacteria (Vibrio sp., Aliivibrio sp. and Bacillus sp.), viruses (Alloherpes virus, Lymphocystis virus and Singapore group iridovirus) and parasites (flagellates, monogeneans, amyloodinium and cryptocaryon) [2].(L 45-49)

M and M:

L 103: How many fishes.

In our present study, 100 healthy clownfish were bred and maintained in the Shenzhen Base of South China Sea Fisheries Research Institute in Guangdong Province, China.

L 113: Clarify total RNA extraction.

We appreciate the helpful reviewer’s comment. Total RNA from clownfish tissues was treated with TRIzol reagent (TransGen, Beijing, China) according to manufacturer's protocol(L 112-114). The aforementioned description had added into the revised manuscript.

L 119: "and cDNA synthesis was performed previously" what does this mean? 

We appreciate the helpful reviewer’s comment and had rephrase the sentence as follow in the revised manuscript.

“and followed by cDNA synthesis (TransGen, Beijing, China) according to manufacturer's protocol as previously described.(L 121-122)

Table 1: -You need to leave a pace between the primer name and F or R.

We appreciate the valuable suggestion and it had been left a pace between the primer name and F or R.

- Correct "Used" in the main raw.

We appreciate the reviewer’s helpful suggestion and had refined the word Used to Purpose”.

L 125-127: Describe more details, it is not enough to just cite your previous work and what does SUPPLEMENT MEAN???

We appreciate the reviewer’s helpful suggestion and had added more details as follow in the revised manuscript.

The cDNA sequence and amino acid sequence of the AoNKlysin similarities were examined via BLAST (http://blast.ncbi.nlm.nih.gov/Blast.cgi). The physical and chemical properties were predicted using ExPASy software (http://www.expasy.org/). The location of domains was predicted by SMART (http://smart.embl-heide lberg.de/). Protein family membership was predicted by the InterProScan program (http://www.ebi.ac.uk/Tools/ pfa/iprsc an/). Protein multiple sequence alignments of AoNKlysin protein was performed by ClustalX 2.0 program with the default parameters and edited by the GeneDoc software. The phylogenetic tree was generated based on the deduced amino acid sequence of AoNKlysin protein with the neighbour‐joining method using MEGA 6 program, in which the Poisson distribution substitution model and bootstrapping procedure with 1,000 bootstraps were applied.(L 127-138)

L 129: "A total of 9 diverse samples were gathered" What does this mean??

We appreciate the reviewer’s helpful suggestion and had added more details as follow in the revised manuscript.

A total of 9 diverse tissue samples were gathered from healthy clownfish.(L 140-141)

L 141: " averaged into 2" WHAT IS THIS??

We appreciate the reviewer’s helpful suggestion. It means that the 100 healthy clownfish were randomly averaged into 2 groups. We artificially injected with 10 μl SGIV (MOI=1) in the experimental group, whereas those in the control group was injected with 10 μl sterilized phosphate-buffered saline (PBS) per fish.

Discussion:

- better to separate results from the discussion.

We appreciate the reviewer’s helpful suggestion. We had separated the results from the discussion in the revised manuscript.

- Try to make a better correlation between your findings.

We appreciate the reviewer’s valuable comment. We had discussed by making a better correlation between our findings in the revised manuscript.

Reviewer 3 Report

In the manuscript, the authors reported the cDNA and amino acid sequence of NK-lysin in Clownfish and compare its expression distribution in different tissues and its expression response during viral infection. Recombinant AoNk-lysin produced in P. pastoris system inhibited the growth of several bacteria and reduced virus RNA levels on cultured cells.

These results seem clear, however, there are some concerns in the manuscript and a few assertions are puzzling.

1.       Overall, there are a great many grammatical problems. I am not a native English speaker, so I understand the difficulty, but I found that many of the problems, such as the position and number of spaces, which are not even English grammar, are never checked and are not up to the level to be peer-reviewed.

2.       The authors wrote in the Abstract that AoNk-lysin had abilities not only for direct pathogen-killing but also as an immuno-enhancement, but it is unclear what immuno-enhancement means. If the term "immuno-enhancement" refers to the ability to suppress viral growth, then more direct evidence should be demonstrated by showing the expression levels of antiviral genes, such as type-I IFN and Mx, in FHM cells upon AoNk-lysin exposure.

3.       In the authors previous study of Nk-lysin in Nile tilapia (ref. 22), the highest expression level of Nk-lysin was in the gill, and the tissue distribution of the OnNk-lysin is not at all consistent with the AoNk-lysin results. While differences among fish species should be considered, these results do not provide evidence that Nk-lysin functions in the same role, and there is insufficient discussion of them. In addition, I wonder that why the authors did not check the AoNk-lysin expression in skin, where the highest expression level was observed, after virus challenging.

4.       On the whole, the novelty of the study is weak, and I do not feel that it has any value beyond the fact that it was researched by Clownfish.

5.       There are numerous problems in the reference part, but there seems to be no value to point out individual corrections anymore. For example, ref. 5 and ref. 28 have no information other than the journal name; ref. 11 has no journal information other than the year of publication; ref. 13, 16, 19, 22, and 27 cite from the same journal, but the form of journal names do not match. These are all very low-level errors.

Author Response

Response to reviewer.

Comments to the Author (in italics) with our responses (bold script)

We believe that we have accommodated the requests of the reviewer and in doing so, the quality of the manuscript has been greatly improved.

Reviewer 3

In the manuscript, the authors reported the cDNA and amino acid sequence of NK-lysin in Clownfish and compare its expression distribution in different tissues and its expression response during viral infection. Recombinant AoNK-lysin produced in P. pastoris system inhibited the growth of several bacteria and reduced virus RNA levels on cultured cells.

These results seem clear, however, there are some concerns in the manuscript and a few assertions are puzzling.

  1. Overall, there are a great many grammatical problems. I am not a native English speaker, so I understand the difficulty, but I found that many of the problems, such as the position and number of spaces, which are not even English grammar, are never checked and are not up to the level to be peer-reviewed.

We appreciate the valuable suggestion. The manuscript has been carefully checked and reviewed by a native English speaker.

  1. The authors wrote in the Abstract that AoNK-lysinhad abilities not only for direct pathogen-killing but also as an immuno-enhancement, but it is unclear what immuno-enhancement means. If the term "immuno-enhancement" refers to the ability to suppress viral growth, then more direct evidence should be demonstrated by showing the expression levels of antiviral genes, such as type-I IFN and Mx, in FHM cells upon AoNK-lysin

We appreciate the valuable suggestion. This description is indeed inappropriate in this article, and we had refined this sentence as follow in the revised manuscript.

The main finding of this study is that AoNK-lysin has the ability to eliminate pathogens and inhibited virus replication. This suggests that AoNK-lysin might act as an essential role in defense against bacteria and virus infection.

  1. In the authors previous study of NK-lysinin Nile tilapia (ref. 22), the highest expression level of NK-lysin was in the gill, and the tissue distribution of the OnNK-lysin is not at all consistent with the AoNK-lysin While differences among fish species should be considered, these results do not provide evidence that NK-lysin functions in the same role, and there is insufficient discussion of them. In addition, I wonder that why the authors did not check the AoNK-lysin expression in skin, where the highest expression level was observed, after virus challenging.

We appreciate the valuable suggestion. In previous study, it was found that NK-lysin in Atlantic salmon (Salmo salar) mucus produced by the skin and might participate in local responses in skin-secreted mucus [1]. Yulema Valeroa et al. [1] found that NK-lysin was contained the higher bacteriostatic activity in skin-secreted mucus. However, Clownfish and sea anemones coexist, and sea anemones have some toxicity. The skin of clown fish release a lot of mucus to protect itself. Combined with the study of Yulema Valeroa, the mucus may contain a large amount of antimicrobial peptides, which needs further study. However, previous studies have shown that the main target of SGIV virus infection is immune tissue [2], not skin, so we did not test it. Of course, your advice gives us some inspiration. Since the amount of NKlysin expression in the clown fish skin is so high, the specific causes of this phenomenon will be further discussed in our future research.

Reference

[1] VALERO Y, CORTÉS J, MERCADO L. NK-lysin from skin-secreted mucus of Atlantic salmon and its potential role in bacteriostatic activity[J/OL]. Fish & Shellfish Immunol., 2019, 87: 410-413. DOI:10.1016/j.fsi.2019.01.034.

[2] QIN Q W, SHI C, GIN K Y H, et al.. Antigenic characterization of a marine fish iridovirus from grouper, Epinephelus spp[J/OL]. Journal of Virological Methods, 2002, 106(1): 89-96. DOI:10.1016/s0166-0934(02)00139-8.

  1. On the whole, the novelty of the study is weak, and I do not feel that it has any value beyond the fact that it was researched by Clownfish.

We appreciate the valuable suggestion. In this study, it was discovered that the AoNK-lysin protein has the ability to effectively suppress the proliferation of various harmful bacteria and the replication of SGIV. Although the evidence available is limited, we think the findings are important and need to be highlighted to aquarium fish as the use of maintenance and reliever therapy becomes more prevalent. In the future, our team will conduct a further study on clownfish disease control and treatment.

  1. There are numerous problems in the reference part, but there seems to be no value to point out individual corrections anymore. For example, ref. 5 and ref. 28 have no information other than the journal name; ref. 11 has no journal information other than the year of publication; ref. 13, 16, 19, 22, and 27 cite from the same journal, but the form of journal names do not match. These are all very low-level errors.

We appreciate the reviewer’s helpful suggestion and the aforementioned references had been unified as follow in the revised manuscript.

  1. Steiner, H.; Hultmark, D.; Engstrm; Bennich, H.; Boman, H.G. Sequence and Specificity of Two Antibacterial Proteins Involved in Insect Immunity. Nature.1981,292, 246–248. https://doi.org/10.1038/292246a0
  2. Yu, D.; Weng, T.; Chen, J.; Lu, Y. Functional Characterization of a Grouper Nklysin with Antibacterial and Antiviral Activity. FishShellfish Immunol. 2022;131, 862-871. doi: 10.1016/j.fsi.2022.10.032.
  3. Acosta, J.; Roa, F.; González-Chavarría, I.; Astuya, A.; Maura, R.; Montesino, R.; Muñoz, C.; Camacho, F.; Saavedra, P.; Valenzuela, A.; et al. In Vitro Immunomodulatory Activities of Peptides Derived from Salmo Salar NK-lysinand Cathelicidin in Fish Cells. Fish Shellfish Immunol. 2019, 88, 587–594, doi:10.1016/j.fsi.2019.03.034.
  4. Lama, R.; Pereiro, P.; Costa, M.M.; Encinar, J.A.; Medina-Gali, R.M.; Pérez, L.; Lamas, J.; Leiro, J.; Figueras, A.; Novoa, B. Turbot (Scophthalmus Maximus) NK-lysinInduces Protection against the Pathogenic Parasite Philasterides Dicentrarchi via Membrane Disruption. FishShellfish Immunol. 2018, 82, 190–199, doi:10.1016/j.fsi.2018.08.004.
  5. Cai, J.; Yu, D.; Wei, S.; Tang, J.; Lu, Y.; Wu, Z.; Qin, Q.; Jian, J. Identification of the Bcl-2 Family Protein Gene BOK from Orange-Spotted Grouper (Epinephelus Coioides) Involved in SGIV Infection. Fish & Shellfish Immunol. 2016, 52, 9–15, doi:10.1016/j.fsi.2016.03.026.
  6. Huang, Y.; Zheng, Q.; Niu, J.; Tang, J.; Wang, B.; Abarike, E.D.; Lu, Y.; Cai, J.; Jian, J. NK-lysinfrom Oreochromis Niloticus Improves Antimicrobial Defence against Bacterial Pathogens. FishShellfish Immunol. 2018, 72, 259–265, doi:10.1016/j.fsi.2017.11.002
  7. Ding, F.F.; Li, C.H.; Chen, J. Molecular Characterization of the NK-lysinin a Teleost Fish, Boleophthalmus Pectinirostris: Antimicrobial Activity and Immunomodulatory Activity on Monocytes/Macrophages. Fish Shellfish Immunol. 2019, 92, 256–264, doi:10.1016/j.fsi.2019.06.021.

Reviewer 4 Report

The present manuscript identified NK-lyisn from clown fish and investigated its activities and potential functions. The manuscript is of interest to researchers working on fish immunology. The authors have addressed the importance of NK-lyisn identification in fish. Overall the manuscript is timely and well written. I have a few minor comments which can be found below.

1. the importance of clown fish to aquaculture must be mentioned in introduction

2. the objectives are clearly defined in the study

3. section 2.5. The author used 2-ddct method to calculate gene expression in healthy tissues. For normalisation, the control tissue must be considered and mentioned in this section

4. section 2.6. How did the Author define the challenge dose 10 μl SGIV solution?

5. section 2.11; Homogeneity of variance must be analysed before ANOVA.

Figure 1c. The details of the figure must be highlighted 

Its better

Author Response

Response to reviewer.

Comments to the Author (in italics) with our responses (bold script)

We believe that we have accommodated the requests of the reviewer and in doing so, the quality of the manuscript has been greatly improved.

Reviewer 4

The present manuscript identified NK-lyisn from clown fish and investigated its activities and potential functions. The manuscript is of interest to researchers working on fish immunology. The authors have addressed the importance of NK-lyisn identification in fish. Overall the manuscript is timely and well written. I have a few minor comments which can be found below.

  1. the importance of clown fish to aquaculture must be mentioned in introduction

We appreciate the reviewer’s helpful suggestion and had added the corresponding description as follow in the revised manuscript.

The development of clownfish to aquaculture increased fishermen's income and declined to catch the wild clownfish in natural sea area, as well as preserving coral reef biodiversity.(Line 43-45)

  1. the objectives are clearly defined in the study

We appreciate the reviewer’s helpful suggestion. The objective had been clearly defined in the revised manuscript.

  1. section 2.5. The author used 2-ddct method to calculate gene expression in healthy tissues. For normalisation, the control tissue must be considered and mentioned in this section

We appreciate the reviewer’s helpful suggestion and the corresponding details had been added in the revised manuscript.

  1. section 2.6. How did the Author define the challenge dose 10 μl SGIV solution?

We appreciate the reviewer’s valuable comment and the challenge dose was chosen in the basis of the previous research on zebra fish [1].

According to the pre-experiment, the 10μl SGIV solution was the most suitable infection volume.

Reference:

[1] RAKUS K, MOJZESZ M, WIDZIOLEK M, et al. Antiviral response of adult zebrafish (Danio rerio) during tilapia lake virus (TiLV) infection [J]. Fish & Shellfish Immunol., 2020, 101: 1-8. DOI:10.1016/j.fsi.2020.03.040.

  1. section 2.11; Homogeneity of variance must be analysed before ANOVA.

We appreciate the reviewer’s helpful suggestion. It is our mistake we forget add this detail about the homogeneity of variance before one way-ANOVA analysis. We had further added it in the revised manuscript.

Round 2

Reviewer 2 Report

The authors have answered most of the comments,  one more thing is that I recommend adding more details about RNA extraction, even if summarized steps i.e. reaction volume, rather than citing the company protocol.

This has been improved in the revised version

Author Response

Response to reviewer.

Comments to the Author (in italics) with our responses (bold script)

We believe that we have accommodated the requests of the reviewer and in doing so, the quality of the manuscript has been greatly improved.

Reviewer 2

The authors have answered most of the comments, one more thing is that I recommend adding more details about RNA extraction, even if summarized steps i.e. reaction volume, rather than citing the company protocol.

We appreciate the valuable suggestion. The detail step of RNA-isolation had been shown as follow in this study.

Total RNA was extracted from various tissues (skin, muscle, eye, intestine , brain and so on) of clownfish and followed by cDNA synthesis (TransGen, Beijing, China) according to manufacturer's protocol as previously described [9]. In short, initial quality and quantity of each RNA sample were checked via an Agilent 2100 Bioanalyzer (Agilent, USA) and their integrities were examined by electrophoresis on 0.8% RNase-free agarose gel. Samples were diluted and RNA concentration was measured via NanoDrop 2000 (Thermo, USA). The same equipment was used to assess purity through the OD260/OD230 nm (2.0–2.4) and OD260/OD280 nm (1.8–2.0) absorbance ratios. The RNA samples with RNA integrity number (RIN) more than 8.0 were considered acceptable for subsequent cDNA synthesis via TransScript One-Step gDNA Removal and cDNA Synthesis SuperMix (TransGen, China) according to the manufacture instruction. Based on the published genome information of clownfish in the NCBI library, the specific primers AoNK-OF/AoNK-OR were used for cloning the ORF sequence of AoNK-lysin (XM-023284512.1) (Table 1).

Reviewer 3 Report

To res.1&5,

The authors only corrected the areas that were pointed out to them, and it seems that they cannot figure out all the corrections without being specifically pointed out to them.

For example,

In line 16, replace the full-width space after “peptide” with a half-width space.

In line 20, replace the full-width space after “length” with a half-width space.

In line 46, insert a space after the period at the end of the sentence.

In line 50, insert a space after the period at the end of the sentence.

In line 54, insert a space after the period at the end of the sentence.

In line 66, insert spaces after “mammals” and “fish”.

In line 88, insert a space after the period at the end of the sentence.

In line 91, replace the full-width space after the period with a half-width space.

In line 113, insert a space after the period at the end of the sentence.

In line 113, insert a space after the period at the end of the sentence.

In line 124 and the Table 1, “AoNk-lysin” should be replaced to “AoNK-lysin”.

In line 126, delete a space before the period and insert a space after the period.

In line 127, 132, and 135, , “AoNKlysin” should be replaced to “AoNK-lysin”.

These are just examples, not all of the mistakes.

I do not believe it is the job of Reviewer to point out every single correction any more.

Although these “minor” mistakes are not part of the academic significance, I believe these are also important and cannot be ignored in presenting the paper.

Likewise in the reference part, the authors only corrected the areas that were pointed out in the previous review comment.

In fact, It is regrettable that the authors are not even aware of any errors that I did not point out as examples in the previous review.

For example,

Ref. 28 have no information other than the journal name (I have pointed put in the previous review).

Ref. 8 and 21 cite from the same journal, but the form of journal names do not match.

Ref. 19, the journal name also should be replace to “Fish Shellfish Immunol.”.

Likewise in ref. 29, the journal name should be rewrite to “Fish Shellfish Immunol.”.

All words that should be italicized should be corrected.

It is a waste of time and very regrettable for both of us. I sincerely hope that no more of these remarks occur.

To res. 3,
I understood the reason why the authors did not check the AoNK-lysin expression level in skin. However, the authors' answer does not seem to be on target. The reported expression patterns of NK-lysin vary widely among fish species, and this diversity should be discussed instead of the relationship with anemones.

Author Response

Response to reviewer.

Comments to the Author (in italics) with our responses (bold script)

We believe that we have accommodated the requests of the reviewer and in doing so, the quality of the manuscript has been greatly improved.

Reviewer3.

To res.1&5,

The authors only corrected the areas that were pointed out to them, and it seems that they cannot figure out all the corrections without being specifically pointed out to them.

For example,

In line 16, replace the full-width space after “peptide” with a half-width space.

・In line 20, replace the full-width space after “length” with a half-width space.

・In line 46, insert a space after the period at the end of the sentence.

・In line 50, insert a space after the period at the end of the sentence.

・In line 54, insert a space after the period at the end of the sentence.

・In line 66, insert spaces after “mammals” and “fish”.

・In line 88, insert a space after the period at the end of the sentence.

・In line 91, replace the full-width space after the period with a half-width space.

・In line 113, insert a space after the period at the end of the sentence.

・In line 124 and the Table 1, “AoNk-lysin” should be replaced to “AoNK-lysin”.

・In line 126, delete a space before the period and insert a space after the period.

・In line 127, 132, and 135, , “AoNKlysin” should be replaced to “AoNK-lysin”.

These are just examples, not all of the mistakes.

I do not believe it is the job of Reviewer to point out every single correction any more.

Although these “minor” mistakes are not part of the academic significance, I believe these are also important and cannot be ignored in presenting the paper.

We appreciate the valuable suggestion. The manuscript has been carefully checked and refined carefully according to reviewer’s suggestion.

Likewise in the reference part, the authors only corrected the areas that were pointed out in the previous review comment.

In fact, It is regrettable that the authors are not even aware of any errors that I did not point out as examples in the previous review.

For example,

・Ref. 28 have no information other than the journal name (I have pointed put in the previous review).

・Ref. 8 and 21 cite from the same journal, but the form of journal names do not match.

・Ref. 19, the journal name also should be replace to “Fish Shellfish Immunol.”.

・Likewise in ref. 29, the journal name should be rewrite to “Fish Shellfish Immunol.”.

・All words that should be italicized should be corrected.

It is a waste of time and very regrettable for both of us. I sincerely hope that no more of these remarks occur.

We appreciate the reviewer’s helpful suggestion and the aforementioned references had been unified in the revised manuscript.

To res. 3,

I understood the reason why the authors did not check the AoNK-lysin expression level in skin. However, the authors' answer does not seem to be on target. The reported expression patterns of NK-lysin vary widely among fish species, and this diversity should be discussed instead of the relationship with anemones.

We appreciate the valuable suggestion. The highest expression of AoNK-lysin in skin was different among the NK-lysin expression in other fishes, while differences among fish species may be considered, especially for the genetic differences. On the other hand, the habitat and living environment may also cause this type of difference. All these descriptions had been mentioned in the revised manuscript.
